# Prospects of Red King Crab Hepatopancreas Processing: Fundamental and Applied Biochemistry

Tatyana Ponomareva, Maria Timchenko , Michael Filippov, Sergey Lapaev and Evgeny Sogorin *

Federal Research Center "Pushchino Scientific Center for Biological Research of the RAS", 142290 Pushchino, Russia; tatyanap91875@gmail.com (T.P.); matimchenko@gmail.com (M.T.); filippmv@gmail.com (M.F.); a3t1v4i1n5@gmail.com (S.L.)
* Correspondence: evgenysogorin@gmail.com; Tel.: +7-915-132-5419

**Abstract:** Since the early 1980s, a large number of studies on enzymes from the red king crab hepatopancreas were conducted. They have been relevant both from a fundamental point of view in terms of studying the enzymes of marine organisms and in terms of rational natural resource management aimed to obtain new valuable products from the processing of crab fishing waste. Most of these works were performed by Russian scientists due to the area and amount of waste of red king crab processing in Russia (or the Soviet Union). However, the close phylogenetic kinship and the similar ecological niches of commercial crab species and the production scale of the catch provide the bases for the successful transfer of experience in the processing of the red king crab hepatopancreas to other commercial crab species caught worldwide. This review describes the value of recycled commercial crab species, discusses processing problems, and suggests possible solutions for these issues. The main emphasis is made on hepatopancreatic enzymes as the most salubrious products of red king crab waste processing.

**Keywords:** commercial crab species; red king crab; waste processing; hepatopancreas; proteases; hyaluronidase

## 1. Introduction

The global growth in consumer demand for food products based on commercial species of marine organisms (fish, crab, squid, etc.) has stimulated fishers to increase production. However, the inability to increase the catch volume endlessly and related waste recycling problems have put forward the consideration of advanced processing of secondary raw materials to obtain new commercially valuable products. Crabs are a favorite catch of fishers worldwide due to the high price of their meat. The main northern commercial species of crab include the following: Red king crab (*Paralithodes camtschaticus*), Blue king crab (*Paralithodes platypus*), Spiny brown king crab (*Paralithodes brevipes*), Golden king crab (*Lithodes aequispinu*), Opilio snow crab (*Chionoecetes opilio*), Tanner snow crab (*Chionoecetes bairdi*), Triangle tanner crab (*Chionoecetes angulatus*), Red snow crab (*Chionoecetes japonicus*), Chinese mitten crab (*Eriocheir sinensis*), Hair crab (*Erimacrus isenbeckii*) [1]. The first four species belong to the infraorder of half-tailed or false crabs (Anomura) of the suborder Pleocyemata, order Decapoda. The rest of them belongs to the infraorder of true crabs (Brachyura) (Figure 1) [2].

All crabs have a massive cephalothorax covered with carapace from above, a flat abdomen, and are bent under the cephalothorax. Brachyura representatives move with the help of four pairs of pectoral legs, and the fifth pair of limbs are claws. A specific feature of the Anomura is the asymmetric structure of the body (the right claw is larger than the left) and the presence of only three pairs of walking legs (one of the five pairs is hidden under the carapace and is used for the regular cleaning of gills) [1,3–5].

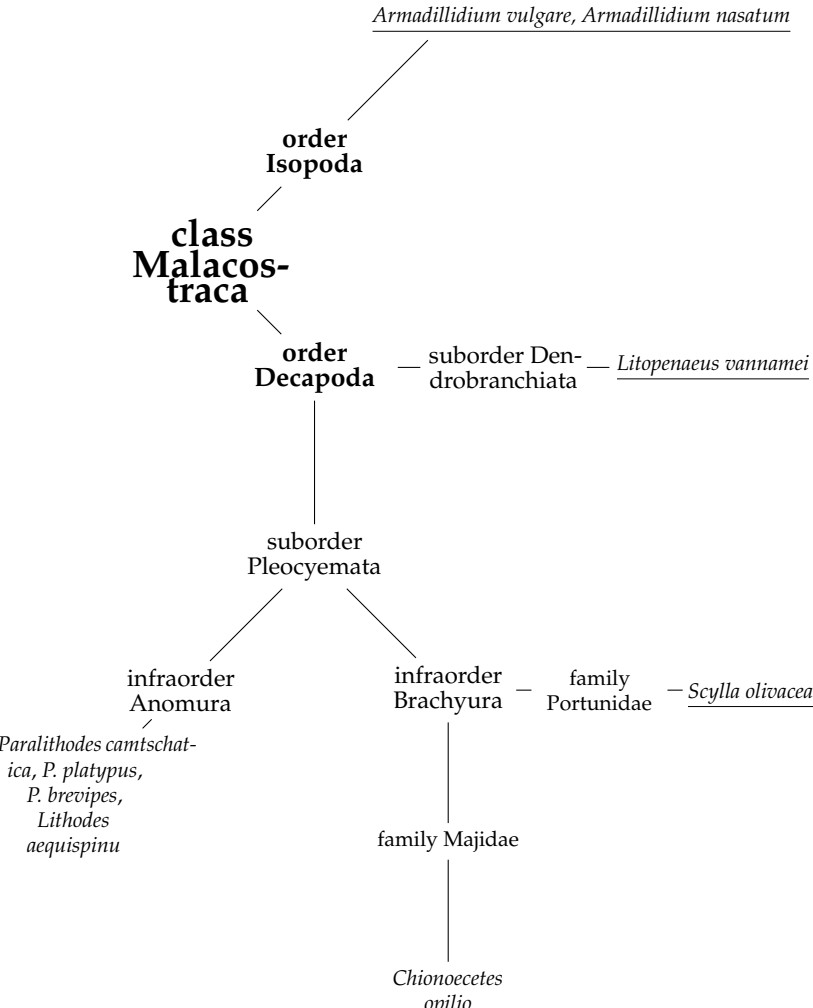

**Figure 1.** Phylogeny of some members of the class Malacostraca. Species with known primary structures of hyaluronidases are underlined (see Section 3.4.3).

Red king crab, or Kamchatka crab, is famous around the world. In recent decades, the catch of this crab by Russian fishers has reached 15,000–20,000 tonnes per year. The original habitat of red king crab ranges from Karaginsky Island (Russia) off the east coast of Kamchatka and Shelikhov Bay (Russia) in the Sea of Okhotsk to Hokkaido Island (Japan) and the Korea Strait (between Korea and Japan). The crab is also found on the west coast of North America from Cape Barrow to the Queen Charlotte Archipelago in the south [1,6].

Red king crab was successfully introduced to the Barents Sea in the 1960s and 1970s. Optimal temperature conditions, the absence of natural predators, and sufficient amount of food led to the spread of acclimatized red king crab from the coast of the Kola Peninsula to Norway and north to Svalbard (Figure 2). Individuals of this king crab population are larger, grow faster, and mature earlier than individuals of the far-eastern population [7]. The rapid growth of the northern king crab population is an environmental problem [7,8]. The growth rate of the resource potential of this population made it possible to open commercial fishing in Norway in 2002 and in Russia in 2004.

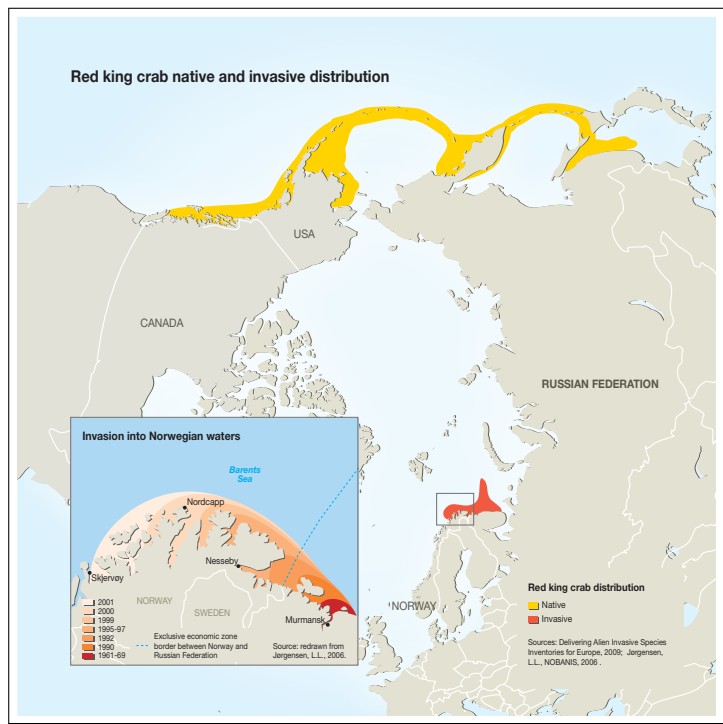

**Figure 2.** Red king crab native and invasive distribution.

In Russia, the crab catch is often processed immediately on board. One of the methods for processing the crab on a ship is as follows: the just-caught crab is placed on a hook; the limbs are additionally cleaned, boiled, and frozen; and the broken carapace with entrails is dumped immediately into the sea as a waste product. On average, this waste can account for 50% of the catch mass [9,10]. The mass fraction of carapace from these wastes is approximately 60%; the rest comprises the entrails (including the digestive organ, the hepatopancreas) [6]. In terms of protein content, these wastes are identical to crab meat, and even superior in terms of the content of minerals, lipids, and carbohydrates. Crab flour used as animal feed can be obtained from crab processing waste. In addition, the crab shell is an excellent source of raw materials for the production of chitin and chitosan, which can be used to meet the demands of the food industry and medicine [10]. Chitin, chitosan, and their derivatives represent promising matrices for the development of novel biomaterials with antioxidant, bactericidal, hypotensive, antiallergic, antiinflammatory and antitumor activities [11]. Recently, crab shell has also been used for the production of $\alpha$-glucosidase inhibitors and anticancer agent prodigiosin via microbial fermentation [12,13]. Due to their peculiar properties such as nontoxicity, biocompatibility, and biodegradability, chitin and chitosan are used as potential excipients and as biological active agents in cosmetology [14].

The main focus of the review is on the research and use of the crab hepatopancreas enzyme complex due to the strong industry interest in this enzyme source. Russia is the leader in the catch of this fishing object. This is reflected in the number of studies on the processing of this waste by Russian scientists. Most of these works were published only in Russian, which significantly limited the availability of information on that research results and development for the world community. The goal of this review is to solve this problem.

## 2. Materials and Methods

The review has been systematically approached based on electronic databases including Scopus, Web of Science, Google Scholar. Most of the resources are the articles of highly specialized journals without translation and abstracts of regional symposia and conferences specialized in the water resources of Russia (Soviet Union), which are not available in electronic form.

Figure "Red king crab native and invasive distribution" was generously provided by GRID-Arendal (nonprofit environmental communications centre based in Norway). This graphic item may be reproduced in any form for educational or nonprofit purposes without special permission from GRID-Arendal, provided acknowledgement of the source is made [15,16].

The chapter "Technologies for processing hepatopancreas" presents patented technologies for processing waste from red king crab and other closely related commercial crabs. This information corresponds to the study of the technical level of patent research on this topic.

Experimental data from research articles were used to compile tables. Among other things, the molecular masses of the proteins under study and the methods used in these works are indicated. The table shows the scatter of these data from different works in order to demonstrate the measurement by different researchers. The correct molecular weight should be estimated by the results of mass spectrometry and interpretation of the results of calculations for the amino acid sequence of proteins.

## 3. Results

### 3.1. Crab Hepatopancreas

The hepatopancreas is an organ of the digestive system that functions as both the liver and pancreas [17]. In red king crab, the hepatopancreas makes up about 90% of the intestines of the carapace and 5–10% of the total weight of the animal [18]. The Decapoda hepatopancreas secretes a wide variety of highly active enzymes.

Under the action of the digestive enzymes of the hepatopancreas, food is broken down into easily digestible substances. The hepatopancreas of crabs of the family Lithodidae (infraorder Anomura, for example, red king crab) consists of brown mass of fragile liver tubules filling the main part of the body cavity. The integrity of these tubules is easily destroyed, even by a slight mechanical action, and the enzymes enter the body cavity, where they start the process of autolysis. The hepatopancreas of true crabs (Brachyura, for example, opilio crab) is shapeless orange-brown mass [1].

### 3.2. Use of Red King Crab Hepatopancreatic Enzymes

The hepatopancreas of the digestive system of commercial crabs is a valuable source of a complex of enzymes with various activities: collagenase, protease, hyaluronidase, lipase, nuclease, etc. The complex of proteolytic enzymes of the red king crab hepatopancreas is of interest in various industries. For example, the prospect of using hepatopancreas enzyme preparation in the hydrolysis of soy protein was recently highlighted [19]. A red king crab hepatopancreas enzyme preparation was successfully used to separate roe from the connective tissue of ovaries of commercial fish [20], to hydrolyze minced fish meat to obtain a dietary food ingredient [21], to hydrolyze crustacean processing waste products to obtain components for microbiological nutrient media [22], and to isolate chondroitin sulfate from marine wastes, namely from tissues of marine organisms [23]. Based on collagenolytic proteinases, wound healing and wound cleansing preparations [24–26], including wound dressings [27,28], have been designed.

### 3.3. Hepatopancreas Recycling Technologies

Many technologies are used to process the hepatopancreas of commercial crab species to obtain enzymatic preparations. Most technologies were developed for the hepatopancreas of red king crab, but some were transferred to the processing of the hepatopancreas from other commercial species (for example, snow crab and blue crab) [29,30]. Most often, the initial raw material is the hepatopancreas (Figure 3), which is homogenized in salt buffers or by osmotic shock under hypotonic conditions (excess distilled water). The integrity of the hepatopancreas tissue is easily destroyed during the freezing/thawing process; therefore, no considerable effort is required for mechanical homogenization. However, in some cases, a colloid mill can be used [31]. Sometimes, triton X-100 or sodium

dodecyl sulfate (SDS) is added to the homogenization buffer [29,32]. The homogenate is incubated at room temperature for several hours. Under such conditions, cell autolysis occurs, which increases the level of protein extraction; however, this can lead to the inactivation of target enzymes. A significant problem in further processing is caused by lipids. Ballast substances and lipids are removed from the homogenate by centrifugation or flocculation followed by chitosan filtration [30–34] or immediately by filtration through the hollow fiber [35,36]. The filtrate is concentrated by ultrafiltration through hollow fibers with a pore size of 15–30 kDa and then dried by freeze-drying or spray-drying. The choice of hollow fibers with a specific pore size (considering the variation in real pore sizes) should be based on the known molecular weights of the target enzymes. To obtain more purified preparations before drying, proteins are precipitated with ammonium sulfate and/or tert-butanol, and then ion exchange, hydrophobic, or affinity chromatography is performed [37–40]. In the simplest case, after homogenization, proteins are precipitated with an excess of cooled acetone ("acetone powder") [41]. This method is not suitable for upscaling, since the use of a large volume of acetone is unsafe for both humans and the environment. Most of the developed technologies have been successfully tested on tens and hundreds of kilograms of hepatopancreas as raw materials [29–31,33–36]. The yield of dry final substance in such protocols varies from 0.6 to 1.3% based on the feedstock [33–36], and the collagenolytic activity is 500–3000 units Mandl/mg (the substrate is collagen type I, at 37 °C) [35,36], even reaching 11,000 Mandl/mg (the substrate is collagen type III, at 42 °C) [29], whereas the amount of protein in dry matter is 80–98% [29,33,35,36].

The basic conditions for enzymes obtained with the mentioned technologies include the maintenance of neutral pH of the enzyme solution; at pH values below 5.5 and above 9.5, the activity of the proteolytic complex and its individual components decreases dramatically. At pH values below 3 and above 10.5, parts of the enzymes are irreversibly inactivated [33,42].

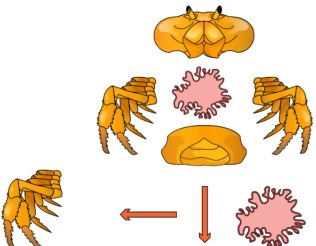

**Homogenization:** freezing/thawing process, osmotic shock, detergent, cell autolysis, or mechanical homogenization

**Ballast substances and lipids removing:** centrifugation, flocculation, or filtration through the hollow fiber

*crude preparation is ready* 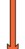

**Additional purification:** precipitation of proteins (ammonium sulfate, acetone, etc.), chromatography



**Figure 3.** Concept of hepatopancreas processing.

### 3.4. Enzymes of the Red King Crab Hepatopancreas

3.4.1. Proteolytic Enzymes

Currently, 10 proteolytic enzymes of hepatopancreas—collagenolytic serine proteinase PC (PC—*Paralithodes camtschaticus*), collagenolytic serine proteinase PC 2, trypsin-like proteinase A, chymotrypsin-like proteinase C, aminopeptidase PC, carboxypeptidase PC, trypsin PC, elastase, cathepsin L, and metalloproteinase—were described in the literature in detail (Table 1). Most of these are small proteins up to 30 kDa (based on the denaturing gel electrophoresis data) with an isoelectric point (pI) of 2.5–4.4. The optimal operating conditions for these enzymes include neutral pH and temperature range from 25 to 55 °C. The collagenolytic serine proteinases are of particular interest due to their ability to degrade native collagen of types I–III. Most likely, these two enzymes are the basis of the active substance in existing wound healing and wound cleaning preparations. Collagenolytic activity has also been assigned to other hepatopancreas proteinases. However, the temperature of the reaction mixture in these studies was often 42 °C, at which native collagen partially denatures [43]; therefore, the measured activity may refer not to true collagenase but to gelatinase activity. Despite the large number of scientific works devoted to hepatopancreas enzymes, the complete nucleotide sequence of mRNA is known only for collagenolytic serine proteinase PC, trypsin PC, cathepsin L, and metalloproteinase (Table 1).

**Table 1.** Hepatopancreas proteolytic enzymes. [1] expected molecular weight based on the mRNA nucleotide sequence, [2] mass spectrometry data. PC—*Paralithodes camtschaticus*.

| Proteinase | kDa (Based on Electrophoresis) | Opt. pH | Opt. °C | pI | Substrate Specificity and Other |
|---|---|---|---|---|---|
| Collagenolytic serine proteinase PC [44] | 29<br>23.5 [1] [45] | 7.5 | 47–55 | 3 | Preferably hydrolyzes peptide bonds, the carbonyl formed by Arg, Lys, and hydrophobic amino acids. Hydrolyzes native collagen type I even at 4 °C [46]. The mRNA nucleotide sequence was determined [45], European Molecular Biology Laboratory (EMBL) Nucleotide Sequence Database: AF461035. |
| Collagenolytic serine proteinase PC 2 [47] | 25 | 8.5 | 38–40 | ? | Preferably hydrolyzes peptide bonds, carbonyl formed by positively charged amino acids. Hydrolyzes native collagen types I–III. |
| Trypsin-like proteinase A [48] | 27<br>30 [49] | 7.9 | 55 [49] | 2.5 | Preferably hydrolyzes peptide bonds, the carbonyl formed by Arg and Lys. Proteolytic activity is not inhibited by ethylenediaminetetraacetic acid (EDTA), and partially inhibited by soybean trypsin inhibitor. |
| Chymotrypsin-like proteinase C [50] | 24 | 9 | 55 [49] | 2.9 [49] | Preferably hydrolyzes peptide bonds, carbonyl formed by hydrophobic amino acids (Phe, Val, and Leu). Not inhibited by Tos-Phe-CH$_2$Cl (chymotrypsin inhibitor). |
| Aminopeptidase PC [51] | 110 | 6 | 36–40 | 4.1 | Effectively cleaves N-terminal amino acids: Arg, Lys, Leu, Phe, and Met. Most likely it is a homodimeric, Zn-containing enzyme. |

<div align="center">**Table 1.** *Cont.*</div>

| Proteinase | kDa (Based on Electrophoresis) | Opt. pH | Opt. °C | pI | Substrate Specificity and Other |
|---|---|---|---|---|---|
| Carboxypeptidase PC [52] | 34 | 6.5 | 55 | 3.1 | Effectively cleaves C-terminal amino acids: Arg, Lys, Phe, Tyr, Leu, and Ile. The enzyme is inhibited by 0.5 mM $Ag^+$, $Zn^{2+}$, $Cd^{2+}$ and 1 mM EDTA, whereas it is activated by $Co^{2+}$ and $Ca^{2+}$. |
| Trypsin PC [53] | 29 <br> 23 [54] <br> 24.8 [1] [45] <br> 24.8 [2] [54] | 7.5–8 | 55 | <2.5 | Preferably hydrolyzes peptide bonds, carbonyl formed by Arg and Lys. The mRNA nucleotide sequence was determined [45], EMBL: AF461036. |
| Elastase [55,56] | 28.5 | 8–8.5 | 50 [57] | 4.5 | Hydrolyzes native elastin (inhibited by elastinal). NaCl, $MnCl_2$, $CdCl_2$ at a concentration of 1–100 mM stimulate elastase activity, whereas it is inhibited by $HgCl_2$ (100 mM). |
| Cathepsin L [58] | 29 <br> 24 [1] | 8 | 25 | ? | Enzyme has cathepsin activity, hydrolyzes Z-Phe-Arg-pNA substrate. Hydrolyzes collagen types X and VI. $HgCl_2$, E64, and leupeptin inhibit cathepsin activity; soybean trypsin inhibitor practically does not suppress activity. The mRNA nucleotide sequence was determined, EMBL: HQ437281 |
| Metalloproteinase [59] | 22.2 [1] | 8–8.5 | 45 | 4.43 | Destroys peptide bonds formed by both acidic and hydrophobic amino acids. Hydrolyzes azocollagen. Proteolytic activity is maintained at 1–3 M NaCl and is inhibited by isopropanol, o-phenanthroline, and EDTA. Zn-containing enzyme. The mRNA nucleotide sequence was determined, EMBL: AF492483 |

### 3.4.2. Nucleases and Other Enzymes of Hepatopancreas

The red king crab hepatopancreas is a source of enzymes such as nucleases (Table 2). $Ca^{2+}$- and $Mg^{2+}$-dependent DNase, which is characterized by high thermal stability, has been studied in detail: the enzyme remains absolutely active after 3 h of incubation at 60 °C, whereas incubation for 30 min at 100 °C resulted in only 7% loss of activity [60]. Based on the circular dichroism (CD) spectrum, the protein appears to be a compact globule and consists mainly of *β*-layers (75%) [61]. The pI value is 4 and the optimal reaction temperature range is 50–60 °C [62].

This DNase has a pronounced specificity for the secondary structure of DNA and predominantly cleaves double-stranded substrates (DNA and RNA–DNA duplexes, while the RNA remains intact). The minimum duplex size for cleavage by DNase is 9 bp; the enzyme does not hydrolyze RNA [62]. The DNase cleaves phosphodiester bonds with the formation of 5′-phosphate and 3′-OH terminal groups. $Ca^{2+}$ and $Mg^{2+}$ together have a positive synergistic effect on the rate of the hydrolysis reaction. The unique properties of DNase enable it to be used effectively for the rapid analysis of single nucleotide polymorphisms

and the normalization of nucleic acid libraries [63–66]. In the hepatopancreas of red king crab, other nucleases (RNases) and phosphoesterases were found (Table 2).

**Table 2.** Nucleases and other enzymes of hepatopancreas. [1] denaturing gel chromatography data, [2] denaturing electrophoresis, [3] expected molecular weight based on the mRNA nucleotide sequence.

| Enzymes | kDa (Based on Gel-Chromatography) | Opt. pH | Known Properties |
|---|---|---|---|
| $Ca^{2+}$- and $Mg^{2+}$-dependent DNase [60] | 53<br>47 [1]<br>42 [2]<br>41.5 [3] [62] | 7–8<br>6.6 [62] | The primary structure was determined. There are two sequences in UniProtKB, Q8I9M9 (2003) and B6ZLK3 (2009), differing by two amino acids. |
| Alkaline RNase (AlkR) [67] | 19 | 7.2–7.5 | Broad specificity. Poorly hydrolyzes poly(AUC). $MgCl_2$ at a concentration of 10–50 mM stimulates the activity of enzyme. 0.1 M NaCl inhibits the enzyme activity by 50%. |
| Acid RNase (AcR and AcR') [67] | 33 and 70 | 5.5 | Does not hydrolyze poly(C) and poly(AUC). $MgCl_2$ inhibits its activity, 0.25 M NaCl inhibits its activity by 50%. Most probably, these are monomeric and dimeric forms of the same protein. |
| Two acidic phosphomonoesterases [68] | 80 and 82 | 5.5 | Do not hydrolyze (3′,5′)cAMP (cyclic adenosine monophosphate); 1.5 M NaCl inhibits the enzyme activity by 20% (protein 80 kDa); 1.1M NaCl inhibits the enzyme activity by 50% (protein 82 kDa) |
| Alkaline phosphomonoesterase [68] | 80 | 7.2–7.5 | Does not hydrolyze (3′,5′)cAMP. 0.4 M NaCl inhibits the enzyme activity by 50%. |
| Acidic phosphodiesterase [68] | 57 | 4.8–5 | 50% inhibition at 1.4 M NaCl. |
| Alkaline phosphodiesterase [68] | 51 | 7.2–7.5 | No inhibition up to 1.4 M NaCl is observed. |

The enzyme preparation from the hepatopancreas contains several other enzymatic activities, which allows this enzyme complex to be used for the depolymerization of *β*-glycosidic bonds in chitosan [69–71]. Lipase activity has also been observed [33]; however, other activities of hepatopancreas enzymes have not been sufficiently studied in comparison with proteolytic and nuclease activities.

### 3.4.3. Hyaluronidase Activity of Hepatopancreas Homogenate

Currently, the structure of hyaluronidases of the Malacostraca class has not been studied in sufficient detail. In the open database of UniProt protein sequences, there are only four representatives of this class for which amino acid sequences of hyaluronidases are available: two representative of Decapods, including Whiteleg shrimp (*Litopenaeus vannamei*, UniProt A0A423SH46) and orange mud crab (*Scylla olivacea*, two UniProt proteins A0A0P4VVV1 and A0A0N7ZAX3), and two representative of Isopoda, which are pill-bug *Armadillidium vulgare* (only one of two proteins is characterized as UniProt A0A444ST78) and *Armadillidium nasatum* (UniProt A0A5N5TJL6) (Figure 1). Earlier, hyaluronidase

activity was found in the complex of enzymes of the red king crab hepatopancreas homogenate [72].

In our previous work, the kinetics of the hydrolysis of hyaluronic acid in cosmetic fillers with the red king crab hepatopancreas homogenate was studied for the first time [73]. Using the methods of turbidimetric analysis, atomic force microscopy, and nuclear magnetic resonance spectroscopy, the kinetics of hydrolysis and the structural transformation of hyaluronic acid under the action of homogenate enzymes were investigated. We found that the obtained homogenate has activity comparable to commercially available hyaluronidase preparations. In this work, we demonstrated the possibility of using an enzymatic preparation based on hepatopancreas homogenate for the treatment of complications after the injection of fillers based on hyaluronic acid. Further studies of the effectiveness and safety of hyaluronidase from hepatopancreas on model animals will enable us to develop new drugs for the treatment of complications of filler injections in the near future.

### 3.5. Other Valuable Non-Protein Components of the Red King Crab Hepatopancreas

The crab hepatopancreas is a source of enzymes and other valuable products. For example, a preparation with the properties of an inhibitor of serine proteinases was obtained and characterized from the hepatopancreas, and its effect on the process of human blood plasma coagulation was studied [74]. The tested inhibitor showed an anticoagulant effect that was more pronounced when combined with heparin. Procedures hwere developed to obtain an inhibitor both from the raw material [75] and in the recombinant form [76].

The hepatopancreas contains a large amount of fat, which varies from 10 to 26% [18,77,78]. In the study of the fractional composition of crab fat, it was found to contain triglycerides at a rate of up to 55%, as well as polyunsaturated fatty acids, including $\omega$-3 (14–24% of the total of all fatty acids). Hepatopancreas fat does not contain toxic substances and can be used as a food supplement or as an ingredient for cosmetic products.

### 4. Discussion

### 4.1. Prospects of Processing Waste from Other Commercial Crab Species

The commercial species of crabs include representatives of the false (Anomura) and true crabs (Brachyura) of Pleocyemata. Brachyura includes opilio crab (*Chionoecetes opilio*), which is also a commercially important catch for marine fisheries. Crabs of this suborder have similar enzymatic activity in their digestive system. For example, the proteolytic activities of enzyme preparations from the hepatopancreas of red king crab and opilio crab are practically the same [21]. The zymogram showed that the hepatopancreas of opilio crab contains at least 10 proteolytic enzymes.

The activity of the proteolytic complex was comparable to the commercially available collagenase of the gas bacillus *Clostridium histolyticum* [79,80]. In these works, proteolytic enzymes of the hepatopancreas of opilio crab, but not red king crab, were isolated and biochemically characterized, and the N-terminal amino acid sequences of proteolytic enzymes were obtained. The authors noted their mistake in their next work [42]. Unfortunately, these incorrect data were published in the UniProt database. For example, the amino acid sequence UniProtKB-P20734 (COGC_PARCM) actually belongs not to *Paralithodes camtschaticus*, but to *Chionoecetes opilio*. The sequence of UniProtKB-P34153 (COG1_CHIOP) and UniProtKB-P34156 (COG4_CHIOP) is not derived from the hepatopancreas proteins of *Chionoecetes opilio*, but from *Paralithodes camtschaticus*. The rest of the N-terminal sequences of proteolytic enzymes of these two crabs in [79,80] completely coincide in the UniProt database, which once again confirms the biochemical similarity of the digestive systems of the representative of Pleocyemata.

The close phylogenetic kinship and similar ecological niches of the commercial crab species, as well as the industrial scale of the catch, provide grounds for the successful transfer of the experience of the processing of the king crab hepatopancreas to other crab species to obtain new valuable products. For example, the enzymatic complex of hepatopancreas of opilio crab was successfully used in the production of protein hydrolysates from cod

waste processing, as well as to improve the consistency and juiciness of cod fillets at the stage of the salting of the semi-finished product [21,30,81].

Russia is a leader in terms of the catch level of red king crab, and this catch is increasing every year (Figure 4a). Most of this catch comes from fishers in the Russian Far East, however, the total catch of Russia and Norway in the Barents and Norwegian Seas is expected in the near future to equal and even exceed the catch level in the Russian Far East [82]. In the United States (Alaska), red king crab is also caught, but to a lesser extent. Fishers in Alaska have caught 200–400 tonnes per year from 2013 to 2020 [83]. However, at the same time, the United States catches about 5000–10,000 tonnes of crabs that are collectively known as king crabs [82]. Other commercial species of crabs are also promising species for advanced processing. For example, Food and Agriculture Organization (FAO) data indicate that the global catch of opilio crab is more than 100,000 tonnes [84] (Figure 4b). We confidently state that high catch levels of large commercially significant crabs, and therefore vast amount of waste, provide an opportunity to develop waste processing technologies and include them in the industrial process. The most promising direction of processing should be considered the processing of the hepatopancreas as a source of new high-margin products.

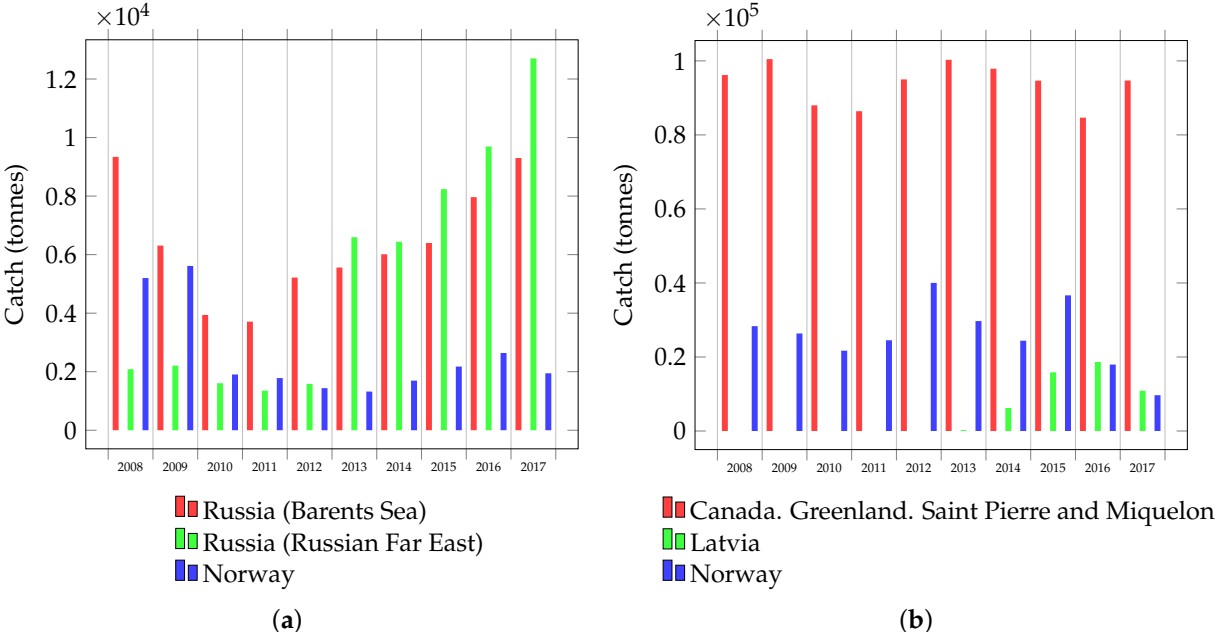

**Figure 4.** Crab catch: (**a**) red king crab in the Barents Sea (Russia and Norway) and the Russian Far East (Russia) and (**b**) the global catch of opilio crab.

### 4.2. Development Strategy for Waste Processing

Crab is processed using different approaches. If the caught crab is not boiled, complex waste processing can be used to maximize the yield of new valuable products. After limb separation, the rest of the body can be completely processed: the carapace can be used for chitosan production, the hepatopancreas for the production of multicomponent enzymatic complexes and specific purified enzymes, the fat from the hepatopancreas for dietary nutrition or biofuel production, and the gills and other internal organs as a feed supplement for birds, fish, and other animals. This approach adopts the principle of non-waste recycling.

The methods of processing the crab hepatopancreas to obtain enzyme preparations can be divided into two strategies: obtaining complexes of various enzymes or further purification of specific enzymes (or class of enzymes) from this complex. The enzyme complex is appropriate for the treatment of multicomponent substrates where the simultaneous degradation of proteins, nucleic acids, and other polymers into monomers is required, for example, in the production of easily digestible food products from animal and plant tissues. The activity of the enzyme complex in the preparations will differ for

the hepatopancreas from sample to sample. In this regard, the technology for using such enzymatic preparations should allow for fluctuations in the activities in different batches of the preparation. The second strategy, based on the isolation of one enzyme from the complex, has advantages, since the final product can be used to produce a high-margin product (for example, a pharmaceutical product or a reagent for scientific research). For this purpose, it is important to have a simple and scalable method for the purification of a specific enzyme. Notably, all the currently existing technologies for processing the crab hepatopancreas do not meet the specified requirements. The main vector of development in this field could be an approach including several stages of tangential flow filtration and affinity chromatography using a cheap carrier.

All these works discussed in this review consider the processing of a specific type of waste (carapace/hepatopancreas/fat). However, the waste is most often a mixture, which is difficult to sort. To implement the principle of non-waste recycling, it is necessary to develop new approaches to recycling this mixture, which will convert the waste into new valuable products. The relevance of deep processing is provided by the sustainable continual increase in catches of fisheries. On the one hand, a growth in catch leads to an increase in waste, which might cause several environmental problems. On the other hand, insufficient control of the catch will lead to a decrease in the crab population, which will entail a sharp decline in the economic performance of the industry in the coming years. Thus, all types of processing could reduce the environmental burden, as well as help satisfy the financial appetites of the industry by selling new valuable products.

The main problem with the complex processing of wastes from commercial marine species is the complexity of their collection and storage under sailing conditions on board a trawler and delivery to coast or further processing. The solution of this problem could be, for example, the creation of a maximally automated integrated waste processing unit immediately on a catching vessel, which could eliminate the problem of the time spent by fishers on the manual removal of an organ, as well as its storage and transportation. Another method of solving this problem is to establish an adequate cost of the hepatopancreas for fishers, allowing a stable supply of raw materials onshore on an industrial scale. The task could be simplified if the non-waste processing of live crabs is organized onshore. Further studies of the hepatopancreas of commercial crab species in fundamental scientific terms considering its potential applications will increase the value of crab processing waste, possibly leading to a time when this will equal or even exceed the cost of crab meat.

## 5. Outlook

Despite a long period of scientific research, deep processing of crab has not been launched yet. The main reason for this is laboratory protocols being not adapted to catch conditions. Also these technologies should provide deep recycling based on the principle of waste-free recycling taking into account financial benefits. It also requires financial support for R&D projects to develop technologies for creating a new high-value products for an industrial scale. The solution to the problem lies in close collaboration between scientists, developers, and fishers. In our review, we discuss all these parties and hope that we touch upon the interests of all the listed stakeholders in order to unite their efforts in the deep processing of wastes from commercial crab species.

**Author Contributions:** T.P. and E.S. conceived of the presented idea, supervised the findings of this work. E.S. designed the figures, and wrote the manuscript with support from T.P., M.T., M.F., and S.L. S.L. developed the theoretical formalism, performed the analytic calculations and performed the statistical data. M.T., M.F., and S.L. processed the literature data, performed the analysis. All authors discussed the results and contributed to the final manuscript. All authors have read and agreed to the published version of the manuscript.

**Funding:** This research received no external funding.

**Institutional Review Board Statement:** At the time of writing this review, no both human or another living thing were harmed (i.e., did not participate in study).

**Informed Consent Statement:** At the time of writing this review, no both human or another living thing were harmed (i.e., did not participate in study).

**Data Availability Statement:** Most of the data is available on internet platforms, some are available only on paper.

**Conflicts of Interest:** The authors declare no conflict of interest. The funders had no role in the design of the study; in the collection, analyses, or interpretation of data; in the writing of the manuscript, or in the decision to publish the results.

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
