# Peer review of "Prospects of Red King Crab Hepatopancreas Processing: Fundamental and Applied Biochemistry"

_recycling, doi:10.3390/recycling6010003_

Round 1

Reviewer 1 Report

The article contains some interesting content that may be valuable for the readers in the field. Thus, I suggest to publish in RECYCLING after consideration of some points as below:

Majors:

  1. “LINE 36-41” should cite the references.
  2. The processing of Red King Crab Hepatopancreas should be also illustrated as a figure contain some steps for clearer for the reader.
  3. “LINE 63” In addition, the crab shell is an excellent source of raw materials for the production of chitin and chitosan, which can be used to meet the demands of the food industry and medicine [6]. In this section, the authors should add some other novel application of crab shell to highlight its values. Such as: “Recently, crab shell has also been used for the production of α-glucosidase Inhibitors and anticancer prodigiosin via microbial fermentation [7,8]

[7]. Nguyen, V.B.; Wang, S.-L. Reclamation of Marine Chitinous Materials for the Production of α-Glucosidase Inhibitors via Microbial Conversion. Mar. Drugs 201715, 350.

[8]. Nguyen, V.B.; Nguyen, D.N.; Nguyen, A.D.; Ngo, V.A.; Ton, T.Q.; Doan, C.T.; Pham, T.P.; Tran, T.P.H.; Wang, S.-L. Utilization of Crab Waste for Cost-Effective Bioproduction of Prodigiosin. Mar. Drugs 202018, 523.

  1. “LINE 65-70” should be moved to the end of the introduction section.
  2. If possible, please add a figure of the Red king crab and Crab Hepatopancreas in section: “3.1”
  3. The titles of all the tables should be put on the top of the table (not at the foot of tables).
  4. “LINE 155-157” should cite the references.

Author Response

We are very grateful to the reviewer for such a thorough review of our manuscript. All comments have been taken into account.  All adjustments undoubtedly improve our work. Please see the attachment.

Reviewer 2 Report

This work has several merits and can be publishable at Recycling. However, many problems must be solved before this as follows: 

*keywords:

I recommend you the maximum of 6 keywords.

  • INTRODUCTION

Lines 17-32: Are there references missing?

Lines 23-32: I recommend you to insert this part immediately after text, without one line for each phrase.

Lines 33-41 and Figure 1: Are there references missing?

Lines 44-45: Could you insert the country of each cited local?

  • Materials and Methods

I believe this section must be included in the introduction section

  • RESULTS AND DISCUSSION

In my opinion, sections 3 and 4 should be assembled and a final section called “outlook” or perspectives must be created talking about the main conclusions and merits of this paper.

Author Response

(The authors gave the same response as above.)

Round 2

Reviewer 2 Report

I believe this paper can be published at Recycling